# Three-Dimensional Point Cloud-Filtering Method Based on Image Segmentation and Absolute Phase Recovery

**Jianmin Zhang [1], Jiale Long [1,*], Zihao Du [1], Yi Ding [1], Yuyang Peng [2] and Jiangtao Xi [3]**

[1] Faculty of Intelligent Manufacturing, Wuyi University, Jiangmen 529020, China; zjm99_2001@126.com (J.Z.); dzher06@gmail.com (Z.D.); dingyi1688@126.com (Y.D.)
[2] School of Computer Science and Engineering, Macau University of Science and Technology, Macau 999078, China; yypeng@must.edu.mo
[3] School of Electrical, Computer and Telecommunications Engineering, University of Wollongong, Wollongong 2522, Australia; jiangtao@uow.edu
[*] Correspondence: longjiale_528@126.com

**Abstract:** In three-dimensional (3D) shape measurement based on fringe projection, various factors can degrade the quality of the point cloud. Existing point cloud filtering methods involve analyzing the geometric relationship between 3D space and point cloud, which poses challenges such as complex calculation and low efficiency. To improve the accuracy and speed of point cloud filtering, this paper proposes a new point cloud filtering method based on image segmentation and the absolute phase for the 3D imaging obtained by fringe projection. Firstly, a two-dimensional (2D) point cloud mapping image is established based on the 3D point cloud obtained from fringe projection. Secondly, threshold segmentation and region growing methods are used to segment the 2D point cloud mapping image, followed by recording and removal of the segmented noise region. Using the relationship between the noise point cloud and the absolute phase noise point in fringe projection, a reference noise-free point is established, and the absolute phase line segment is restored to obtain the absolute phase of the noise-free point. Finally, a new 2D point cloud mapping image is reconstructed in 3D space to obtain a point cloud with noise removed. Experimental results show that the point cloud denoising accuracy calculated by this method can reach up to 99.974%, and the running time is 0.954 s. The proposed method can effectively remove point cloud noise and avoid complex calculations in 3D space. This method can not only remove the noise of the 3D point cloud but also can restore the partly removed noise point cloud into a noise-free 3D point cloud, which can improve the accuracy of the 3D point cloud.

**Keywords:** fringe projection profilometry; point cloud filtering; 3D shape measurement; 3D data processing

## 1. Introduction

Three-dimensional (3D) imaging is the technology fusing sensors and computing methods together to obtain the 3D surface profile of a target object [1]. With the development of image processing technology, sensor technology, and optical technology, 3D imaging technology is widely used in various fields [2,3]. At present, the main methods involved in non-contact 3D imaging technology are the binocular stereo vision method [4,5], the photometric stereo method [6,7], structured light 3D imaging [8–10], etc. The binocular stereo vision method [11] is the principle of binocular visual observation by simulating human beings. This method uses corresponding binocular cameras to take images of objects at different angles and uses image processing technology to calculate the offset and matching between image pixels to obtain three-dimensional information. The photometric stereo method [6] is an image-based 3D imaging method that utilizes multiple images captured from different angles or under different lighting conditions to obtain the three-dimensional shape information of a target object. It exploits visual features such as brightness, color,

and shading of light to infer the object's depth and geometric structure by computing the differences or matches between the images, thereby enabling 3D reconstruction. 3D imaging technology based on fringe projection [12–14] is a three-dimensional imaging technology that reproduces the reality of measured objects by using patterns with special structures (such as discrete light spots, streak lights, and coded structured lights). However, in the process of 3D measurement, it will be affected by the accuracy of the instrument, environmental changes, and the structure of the measured object itself, which will inevitably generate noisy point clouds. It will affect the accuracy of 3D measurement [15–17]. Removing point cloud noise to obtain a complete 3D point cloud model has always been the research focus of 3D measurement.

To improve the accuracy of 3D point clouds, various researchers have proposed point cloud denoising methods. Galea et al. [18] introduced a light field-based approach using uncertainty measures and geometric and photometric properties to estimate depth values and correct outliers. This method suffers from inaccuracies in correcting point locations for complex light field data, which introduce errors in the point cloud during denoising. Leal et al. [19] combined median filtering and sparse regularisation to preserve sharp features in point cloud models affected by Gaussian and impulse noise. It needs to adjust parameters for specific scenes and complex point cloud data, and it requires a lot of computing resources when processing large-scale point cloud data. Luo and Hu [20] proposed a score-based method to evaluate noise levels and apply denoising strategies and filters, but it requires more computing resources when dealing with large-scale point cloud data. Irfan and Magli [21] utilised graph wavelets to jointly denoise geometric and colour attributes in point clouds. It has disadvantages in terms of long computation times, high computing resource demands due to the spectrogram wavelet transform, and limitations in handling specific scenes and complex point cloud data. Mao et al. [22] presented a deep learning-based framework called PD-Flow, achieving high-precision denoising through normalised flow and noise separation. This method requires a large amount of training data and computing resources based on the deep learning model. For different types of point cloud data, it is necessary to adjust the network structure and parameter settings.

In 3D reconstruction with fringe projections, absolute phase recovery is a critical process that converts the wrapped phase into the continuous phase [23,24]. The characteristic of the absolute phase of fringe projection makes the structural features of 3D point cloud data, so it is more suitable for subsequent data analysis and processing [25–27]. If the property of absolute phase can be used to remove the influence of point cloud noise, it is expected to improve the quality of 3D point cloud with efficiency and accuracy. It is also possible to recover the true value of some shadow areas with the property of absolute phase, which helps to obtain the whole 3D model.

Based on this consideration, we propose a method to obtain the 3D point cloud data from fringe projection with improved speed and quality. This method is developed based on point cloud mapping image processing. The mapping image is segmented and filtered, and the absolute phase of the shadow area is restored to obtain a highly accurate 3D point cloud. The proposed method can eliminate the noise in the point cloud by restoring the absolute phase and preserving the geometric details of the object. The proposed method can simplify point cloud processing and improve computational efficiency, which provides an effective and efficient solution for point cloud processing and application of the 3D imaging results of fringe projection.

The remaining sections of this paper are organized as follows. Section 2 presents the proposed method of this paper. Section 3 shows the experimental results, result analysis, and discussion. Section 4 concludes the whole paper.

## 2. Principle

### 2.1. Point Cloud Noise Analysis

The three-dimensional measurement method based on fringe projection casts multiple groups of fringes with different periods onto the object; the camera captures the

deformed fringe patterns and performs the fringe analysis to obtain the wrapped phase image with Phase Shift Profilometry (PSP) [28,29]. The phase unwrapping algorithm was employed to unwrap the phase map to obtain the absolute phase map, as shown in Figure 1. Combining the system calibration parameters and absolute phase diagram in Figure 1 for calculation, the three-dimensional coordinates of the object to be measured are obtained for three-dimensional reconstruction. During the three-dimensional calculation process, the relationship between the $Z$ coordinate in the three-dimensional coordinates $(X, Y, Z)$ and the pixel coordinates $(r, c)$ of the absolute phase map is shown in Equation (1):

$$\lambda_c \begin{bmatrix} r \\ c \\ 1 \end{bmatrix} = \begin{bmatrix} \frac{-f}{dx} & 0 & r_0 \\ 0 & \frac{-f}{dy} & c_0 \\ 0 & 0 & 1 \end{bmatrix} \begin{bmatrix} R_c & T_c \end{bmatrix} \begin{bmatrix} X \\ Y \\ Z \\ 1 \end{bmatrix} \tag{1}$$

where $\lambda_c$ is the scaling factor, $f$ is the focal length of the camera, $\begin{bmatrix} R_c & T_c \end{bmatrix}$ is the external parameter matrix of the camera, which is determined by the system calibration parameters; $dx$ and $dy$ are the physical lengths corresponding to $r$ direction and $c$ direction, respectively, $r_0$ and $c_0$ are the physical coordinates of the image pixels in the camera, respectively. Therefore, the point cloud mapping image that needs to be established contains both the position information of the absolute phase map and the information of the three-dimensional point $(X, Y, Z)$.

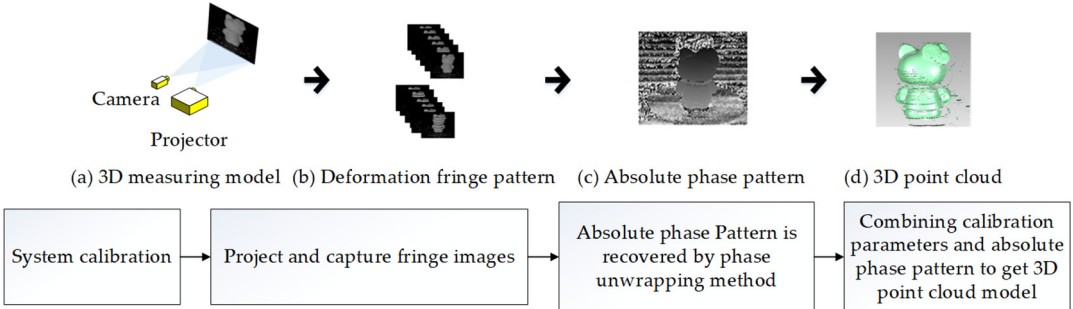

(a) 3D measuring model    (b) Deformation fringe pattern    (c) Absolute phase pattern    (d) 3D point cloud

**Figure 1.** Schematic diagram of 3D imaging based on fringe projection.

Due to factors such as the external environment, measuring equipment, 3D reconstruction algorithms, and the object's own points, the noise will inevitably be generated during the 3D reconstruction of the object. As shown in Figure 1d, the reconstructed 3D point cloud model has a large number of scattered, blocky, and noisy point clouds. The types of noise point clouds produced by 3D reconstruction include scattered noise point clouds, blocky noise point clouds, sudden noise point clouds, etc. [30–32]. According to Equation (1), it is concluded that there is a corresponding relationship between the noise points of the 3D point cloud and the point cloud mapping image, and the noise point cloud can be removed by using the corresponding relationship.

*2.2. Basic Framework*

The point cloud filtering method based on image segmentation is to calculate the corresponding point cloud mapping image and judges the noise region through image segmentation technology. The noise region judged above is deleted to remove the point cloud with noise. Part of the blank region within the outline of the object is restored to obtain a noise-free 3D point cloud.

The schematic diagram of the proposed method is shown in Figure 2, and the overall process of implementation is as follows:

- The 3D point cloud is obtained by 3D reconstruction of the object in Figure 2a, and the point cloud mapping image is established based on the 3D point cloud in Figure 2b, as shown in Figure 2c. The point cloud mapping image is a binarized image;

- Image segmentation is performed on the point cloud mapping image, as shown in Figure 2d. Calculate the area of each region and the total region in the point cloud mapping image to judge the noise region and the noise-free region, and then further judge the non-noise region to obtain the reference region without noise and the undetermined region that may have noise, as shown in Figure 2e. The point cloud mapping image in Figure 2f is obtained by removing the judged noise region, which only contains the reference region and the noise region;

- Determine whether the undetermined region contains noise, fill holes in the reference region, and use the K-nearest neighbor (KNN) algorithm to further judge the undetermined region and the reference region after hole filling to obtain the undetermined region containing noise, as shown in Figure 2g. Calculate the distance between the contour point of the undetermined region of the nearest neighbor point and the three-dimensional point Z of the contour point of the reference region. If the threshold value greater than the set distance is judged as a noise point, the noise region in the image is deleted to obtain Figure 2h;

- Restore the point cloud. According to the point cloud mapping image in Figure 2h, absolute phase image, and three-dimensional calibration parameters, the three-dimensional calculation is performed to obtain a three-dimensional point cloud with noise removed. Based on the denoised 3D point cloud, the point cloud mapping binary image is established, and the point cloud mapping binary image in Figure 2c is XORed within the range of the object contour to obtain the point cloud mapping image of the restored region. The point cloud mapping image is 3D reconstructed and added to the denoised 3D point cloud model to obtain the restored 3D point cloud model.

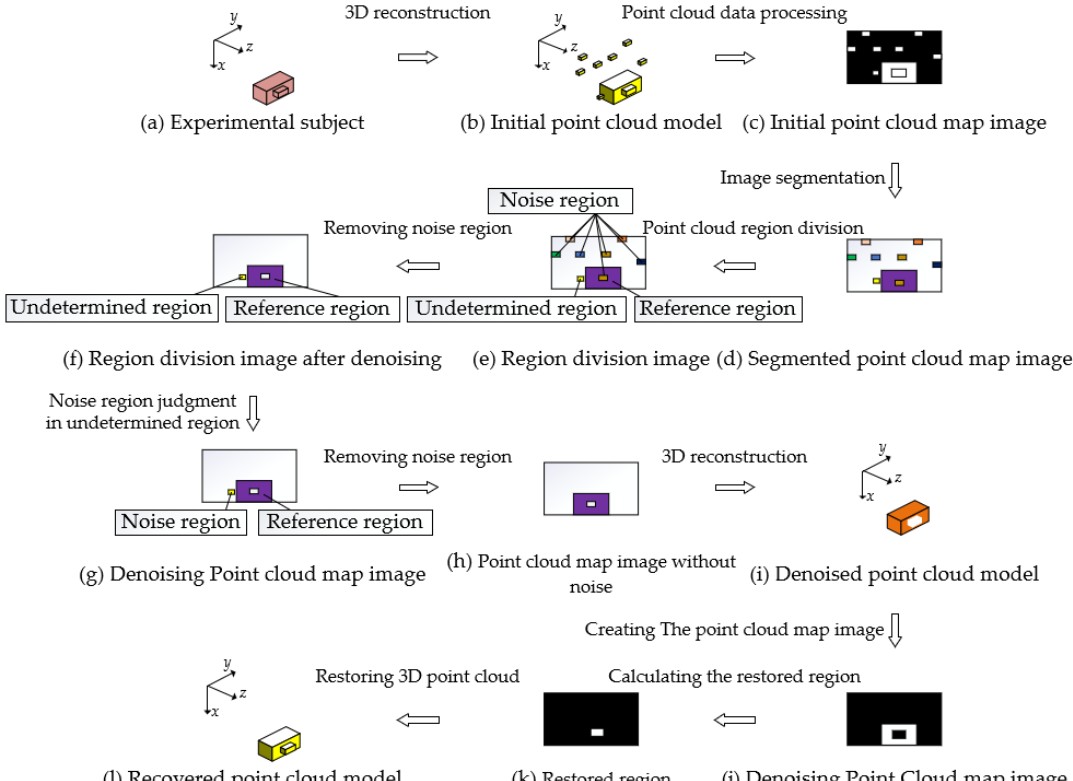

**Figure 2.** Schematic diagram of 3D point cloud filtering method based on image segmentation.

### 2.3. Point Cloud Mapping Image Creation

Figure 3 is the Schematic diagram of the point cloud mapping image creation process. In the process of establishing a point cloud mapping image, calculate the minimum value $x_{\min}$ and $y_{\min}$ of the point cloud $x$ and $y$ direction coordinates, subtract the corresponding minimum value from $x$ and $y$ direction values of the point cloud to obtain new point cloud

coordinates, and convert the point cloud coordinate data into integer data, number each point, map the new point cloud coordinate value to the image, and mark the point cloud serial number and point cloud quantity corresponding to the coordinate point, as shown in Figure 3a,b.

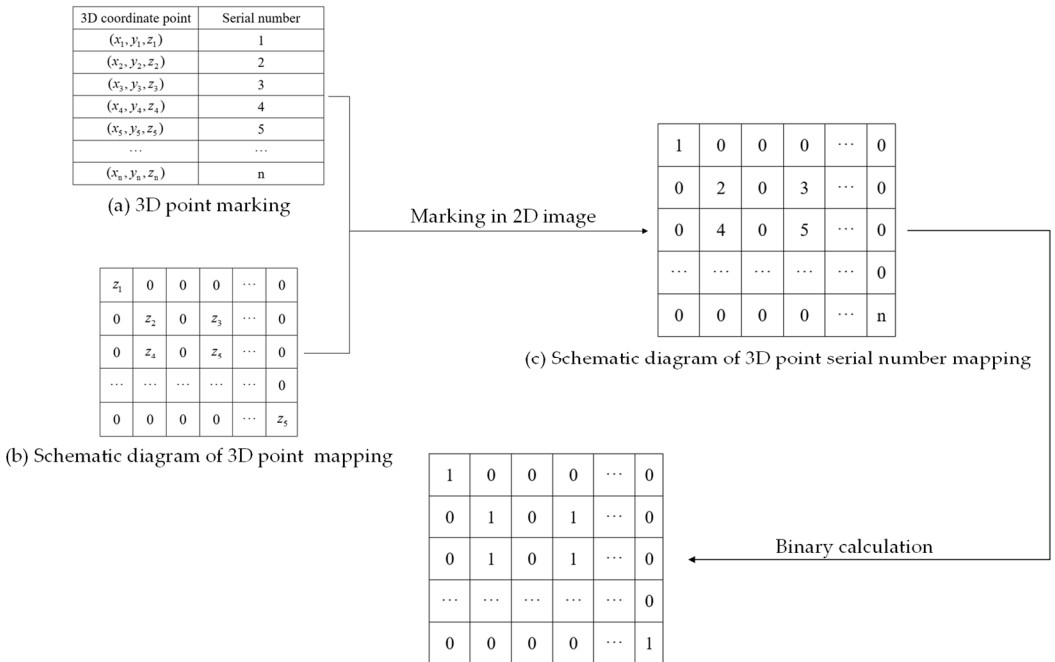

**Figure 3.** Schematic diagram of point cloud mapping image creation process.

The point cloud serial number mapping image in Figure 3b reflects the position and sequence information of the point cloud on the two-dimensional plane. In order to facilitate subsequent processing, the image needs to be binarized. The specific steps are as follows: First, the image coordinates of the serial number are extracted, the corresponding pixel points are assigned a value of 1, and the pixel points corresponding to all image coordinates are assigned a value of 0, and a binary image of point cloud mapping is generated, as shown in Figure 3c. The image binarization in this step is to convert the grayscale image into a black-and-white binary image so that the target area and the background area in the image are more clearly defined. The binarized image mapped by the point cloud can be conveniently used for point cloud segmentation, denoising, and other operations. It should be noted that in the binary image of point cloud mapping, the serial number information has been discarded, and only the position information of the point cloud on the two-dimensional plane is retained. Therefore, if the serial number information of the point cloud needs to be used in subsequent processing, the serial number information of the point cloud should be saved before the binarization process so as to facilitate subsequent processing and analysis.

### 2.4. Mapping Image Segmentation

Perform image segmentation on the point cloud mapping image, set the growth region threshold $\omega = 10$ before image segmentation, and use the region growing image method [14,15] for region segmentation. Select the initial point as the seed point for region growth to segment the point cloud mapping image. The initial point $(x_0, y_0)$ is the point

where the pixel $I(x, y)$ in the point cloud mapping image is not 0, and the coordinates $(x, y)$ are the smallest, which can be expressed as

$$\begin{cases} I(x_0, y_0) \neq 0 \\ x_0 = x_{\min} \\ y_0 = y_{\min} \end{cases} \tag{2}$$

where $x_{\min}$ is the minimum value of $x$, $y_{\min}$ is minimum value of $y$, $I(x_0, y_0)$ is the pixel value of the initial point $(x_0, y_0)$.

If the absolute value of the difference between the initial growth point $(x_0, y_0)$ and the eight neighboring pixels of the point is less than or equal to the regional growth threshold, then the point can be regarded as a point that satisfies the region growing criterion. The neighboring pixel points that satisfy the region growing criterion are the points of the growth region, and the point is recorded. In the next region growing, set all the points in the previous growth region as the new initial growth point, and judge the growth criterion on its eight neighboring pixels. If neighboring pixel points satisfy the region growing criterion, they will be the points of the growing region. Until all the growing points do not satisfy the growth criterion, the growing region stops growing, and this region is divided into a region of image segmentation. Delete the segmented region in the point cloud mapping image, and continue to segment the remaining regions of the image. The cycle repeats until all regions are divided. Each time the regions are divided, the number of growth points is recorded, the region is divided according to the number of growth points, and RGB values are randomly generated and displayed on the point cloud mapping image in Figure 2d. The calculation process of the area of the region is as follows:

$$\begin{cases} S_i = numbel(\text{region}(i)) \\ S = \sum\limits_{i=1}^{n} (S_i) \end{cases} \tag{3}$$

where region(i) represents the i-th segmented region, *numbel* is the calculation of the number of elements in the area, $S_i$ is the area of region(i), $S$ is the total area of all the regions.

The noise region and noise-free region are judged by calculating the area of each region and the total area of all the regions. When region(i) is smaller than $0.1 * S$, the region is a noise region; otherwise, it is a noise-free region. The noise-free regions can be further judged to obtain the reference regions of the noise-free point cloud and the undetermined regions where noise may exist. When region(i) of noise-free regions is larger than 5000 pixels, the region is a reference region; otherwise, it is an undetermined region. The results of all region divisions are shown in Figure 2e.

*2.5. Noise Regions Judgment*

Delete the noise region in Figure 2e, as shown in Figure 2f. Next, determine whether the undetermined regions contain noise regions, fill holes in reference regions, and use the KNN algorithm [33,34] to further judge the undetermined regions and the reference regions after hole filling to obtain the noise undetermined region, as shown in Figure 2g. Calculate the value of the contour points of the undetermined regions of the nearest neighbor points and the contour points of reference regions on the depth value of the 3D point. The calculation process is as follows:

$$D = \sqrt{(Z_1 - Z_2)^2} \tag{4}$$

where $Z_1$ and $Z_2$ are respectively the depth value of the pixel point in the undetermined region of the nearest neighbor point and the depth value of the pixel point in the reference region. $Z_1$ and $Z_2$ can be calculated through the initial three-dimensional reconstruction. $D$ is the Euclidean distance between $Z_1$ and $Z_2$.

Set $\delta$ as the threshold of the Euclidean distance. If $D > \delta$, the undetermined region is the noise region, otherwise, the undetermined region is the reference region. In the process of judging the noise region, the nearest neighbor points of all reference regions and undetermined regions are calculated, and all noise regions in the undetermined region are judged, as shown in Figure 2g. The noise region in Figure 2g is deleted to obtain the denoised point cloud mapping image, as shown in Figure 2h. According to the denoised point cloud mapping image and the phase image calculated in the interference fringe three-dimensional imaging process, the new phase map is obtained by matrix point product calculation. The phase image is combined with the original 3D calibration parameters for the 3D calculation to obtain a noise-free 3D point cloud model, as shown in Figure 2i.

*2.6. Point Cloud Restoration*

The denoised 3D point cloud is restored to a noise-free point cloud model. Create a point cloud map image from the noise-free point cloud model, as shown in Figure 2j. Perform an AND operation on the two images outside the noise-free point cloud region so as to eliminate the influence of other scattered and blocky noise regions on the image that needs to be restored to the point cloud mapping, and perform an XOR operation in the noise-free point cloud region, The region where the pixel values of the two mapping images are different in the point cloud region is obtained, which is the region of the point cloud that needs to be restored. If the pixel values of the two mapping images are both 1 in the cloud region without noise points, the XOR operation can be used to obtain that the pixel values in this region are all 0; that is, there is no point cloud to be restored. Because each noise point corresponds to a unique serial number, map the image through the 3D point cloud serial number, find its corresponding coordinate points in the 2D absolute phase image, and mark these coordinate points, as shown in Figure 2k. There is a one-to-one correspondence between the coordinate point of the absolute phase map and the index of the coordinate point of the three-dimensional point cloud serial number mapping image, and the noise region is restored according to the index of the noise point in the absolute phase map. In the absolute phase diagram containing noise points, the start point $(x_1, y_1)$ and end point $(x_n, y_n)$ of each noise point are searched and set. Set the point $(x_0, y_0)$ before the start point of the noise point and the point $(x_{n+1}, y_{n+1})$ after the endpoint, where the points $(x_0, y_0)$ and $(x_{n+1}, y_{n+1})$ are absolute Noise-free points on phase. According to the points $(x_0, y_0)$ and $(x_{n+1}, y_{n+1})$, calculate the slope $a$ and intercept $b$ of the line segment connecting two points:

$$\begin{cases} a = \frac{y_{n+1} - y_0}{x_{n+1} - x_0} \\ b = y_0 - ax_0 \end{cases} \tag{5}$$

According to the slope and intercept calculated by Equation (5), establish a reference line segment of the noise-free point with the abscissa of the noise point and define the value of the line segment as the reference absolute phase line segment; it can be written as

$$\Phi_r = y = ax_i + b \tag{6}$$

where $\Phi_r$ is the reference absolute phase.

The absolute phase, including noise points, can be calculated as

$$\Phi_e = \phi + 2\pi m_e \tag{7}$$

where $\phi$ is the wrapped phase, and $m_e$ is the fringe order of the calculation error. $\Phi_e$ is the calculated absolute phase. In the 3D reconstruction based on fringe projection, the noise point cloud generated is because the calculation of the fringe order $m_e$ is wrong in the previous process of recovering the absolute phase, the absolute phase $\Phi_e$ obtained is wrong, and the wrapped phase $\phi$ itself is correct.

Recalculate the fringe order *m* corresponding to the noise point according to the reference absolute phase $\Phi_r$, it can be written as

$$m = \frac{\Phi_r - \Phi_e}{2\pi} \qquad (8)$$

Recalculate the absolute phase point corresponding to the noise point according to the new fringe order and Equation (8):

$$\Phi = \phi + 2\pi m \qquad (9)$$

The absolute phase noise recovery is accomplished by calculating the noise-free absolute phase $\Phi$. Figure 4 is a schematic diagram of the absolute phase recovery process, the curves of the phase map are shown in blue. Figure 4a is an absolute phase diagram including noise. According to the judgment of the noise region, after the noise region in Figure 4a is removed, the starting and ending points of the noise region are recorded. Calculate the original noise region according to the above steps to obtain the reference absolute phase line segment and mark the reference line segment with a red line segment in Figure 4b. The new absolute phase is calculated by combining the reference phase line segment and Equations (8) and (9) and connected with the original line segment that does not contain the noise region to obtain the restored absolute phase information, as shown in Figure 4c. According to Figure 1, combined with the original 3D calibration parameters and the new noise-free absolute phase $\Phi$, the 3D reconstruction of the new absolute phase image is performed to obtain a noise-free 3D point cloud, as shown in Figure 2l.

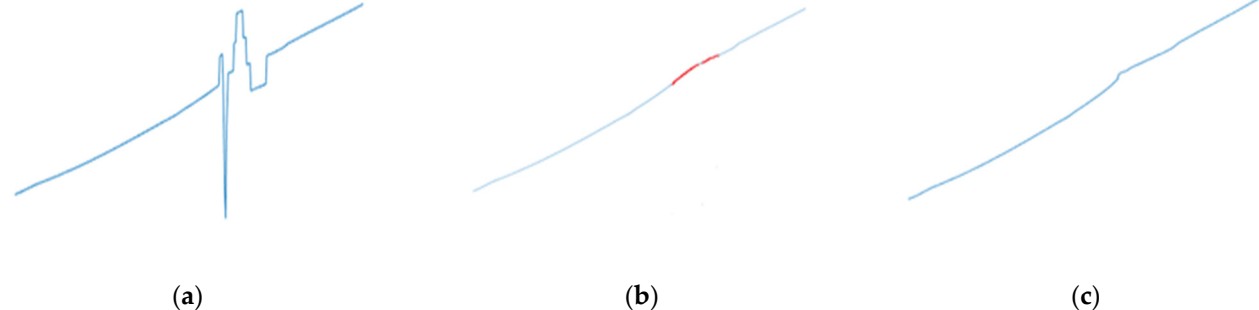

(**a**)  (**b**)  (**c**)

**Figure 4.** Schematic diagram of absolute phase recovery process. (**a**) Initial phase. (**b**) Reference phase. (**c**) Recovery phase.

## 3. Experiments and Discussion

### 3.1. 3D Point Cloud Data Acquisition

In order to verify the effectiveness of the proposed method, after the three-dimensional measurement system is calibrated according to Figure 1, a Hello Kitty doll is selected as object 1 to be measured in the three-dimensional reconstruction experiment. This 3D measurement experiment uses a Texas Instruments (TI) DLPLCR4500EVM projector with a resolution of $1280 \times 800$ pixels and a Basler ace industrial camera with a resolution of $1280 \times 720$ pixels. The proposed method is implemented using MATLAB2020b software and tested on a computer (16 GB Random Access Memory, CPU Intel i5-9300 2.40 GHz main frequency, GPU GTX1650 4 GB). A doll is selected as the experimental object, as shown in Figure 5a, and one of the deformed fringe images captured by the camera is shown in Figure 5b. According to the three-dimensional topography measurement principle based on dual-wavelength fringe projection in Figure 1, the absolute phase map is obtained by analyzing and phase unwrapping multiple deformed fringe patterns by PSP, as shown in Figure 5c. Combining the 3D calibration parameters and the absolute phase map of the object for 3D reconstruction, the reconstructed 3D point cloud is shown in Figure 5d. According to the results of 3D reconstruction, there are scattered and blocky noise point

clouds around the 3D point cloud of the object, and it is necessary to perform a point cloud filtering operation on the 3D point cloud.

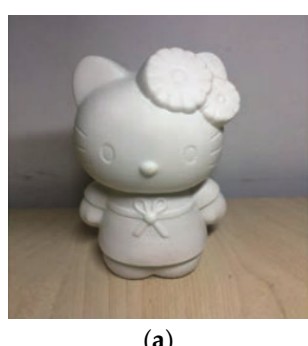 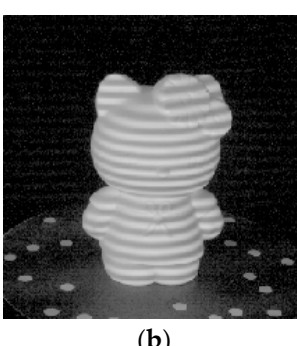 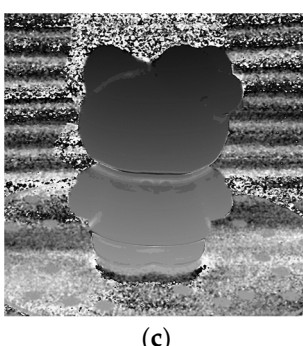 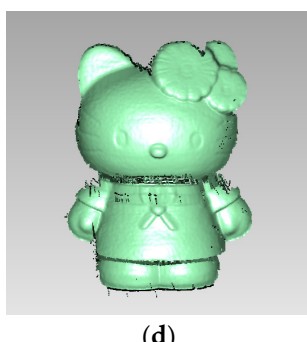

| (**a**) | (**b**) | (**c**) | (**d**) |

**Figure 5.** Three-dimensional reconstruction experiment image of Object 1. (**a**) Experimental object 1. (**b**) Deformed fringe image. (**c**) Absolute phase pattern. (**d**) Three-dimensional point cloud of object 1.

*3.2. Point Cloud Denoising Performance Evaluation Indicators*

In the following point cloud denoising experiment, this paper uses two indicators, namely point cloud denoising accuracy and point cloud denoising time, as the basis for evaluating the performance of point cloud denoising. In this section, a brief description of the evaluation scale used is given. The point cloud denoising accuracy is an evaluation index calculated based on the initial point cloud and the restored point cloud results. It can be written as:

$$Q = \frac{P_1}{P_1 + P_2} \times 100\% \tag{10}$$

where $P_1$ is the number of point clouds in the reference regions determined after mapping image segmentation and KNN calculation. $P_2$ is the number of recovery area of point clouds. If the value of the point cloud denoising accuracy is close to 1, it means that the point cloud denoising performance is good.

The point cloud denoising time is the time for point cloud denoising calculation after the noise point cloud is obtained, including the calculation time for determining all noise regions for removal and phase recovery. If the point cloud denoising time is less, it proves that the point cloud denoising method used is fast.

*3.3. Point Cloud Denoising Experiment 1*

A point cloud denoising experiment based on image segmentation is carried out on the object1. The image of the point cloud denoising experiment is shown in Figure 6. According to Section 3.2, image segmentation, the point cloud map in Figure 6a is segmented into regions. Set the initial point, that is, the growth region seed point $(x_0, y_0)$; the window size of the region growth is 3 pixels × 3 pixels, the step size of the region growth is 1 pixel, and the threshold of each region growth threshold is $\omega = 15$. A schematic diagram of the region's growing process is shown in Figure 7. The table in Figure 7 is part of the point cloud mapping image pixel values; the initial point is (717,499). In the first region growing, (716,500) in 8 neighbor pixel points of (717,499) only can satisfy the region growing principle. (716,500) is the point of the growing region. (716,500) is the judgment point for the next region growing. In the second region growing, only (715,501) in eight neighbor pixel points satisfies the region growing criterion and is a pixel in the growing region, then (715,501) is also a pixel in the growing region. In the third region growing, there are four pixels that meet the region growing criterion and are eight neighbor pixels of (715,501), respectively (714,501), (714,502), (715,502), (716,502). According to the above steps, the regional growth is continued for the four pixels until all the growth points do not satisfy the calculation of the region growth criterion, then, the regional growth is stopped, and the region is recorded. Calculate the area of the region according to Equation (4), and the area of the region is $S_1 = 46529$. The recorded region is shown in Figure 8a, and the area is deleted in the

point cloud mapping image, as shown in Figure 8b. Continue to select the initial points that satisfy Equation (3) in the remaining regions to be segmented as the seed points for region growth until all non-zero pixel points in the figure are in each segmented region. After the image is segmented, use different RGB values to represent each segmented area on the point cloud mapping image. In the KNN algorithm, the parameter settings are $k = 1$ and Euclidean distance threshold $\delta = 15$. We select the nearest neighbor point in the reference region and the undetermined region, respectively, to calculate the Euclidean distance between the two points. The calculated distance is compared with D to determine whether the undetermined region is a noise region.

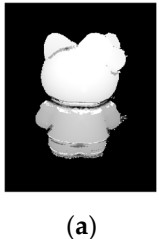

(a)

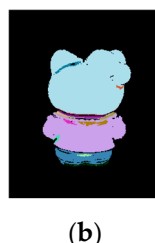

(b)

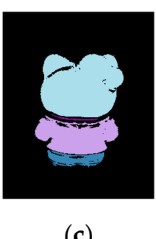

(c)

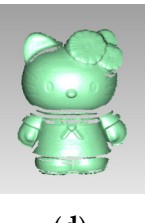

(d)

**Figure 6.** Experimental image of point cloud denoising method based on image segmentation of object 1. (**a**) Point cloud mapping image. (**b**) Segmented mapping image. (**c**) Denoised image. (**d**) Denoised 3D point cloud.

1024x1280 double

|     | 498 | 499 | 500 | 501 | 502 | 503 | 504 | 505 |
|-----|-----|-----|-----|-----|-----|-----|-----|-----|
| 701 | 0 | 0 | 80.8181 | 149.5000 | 150.0936 | 150.7586 | 151.2996 | 151.8844 |
| 702 | 0 | 0 | 148.8810 | 149.5509 | 150.1119 | 150.7549 | 151.2993 | 151.8875 |
| 703 | 0 | 0 | 0 | 149.4784 | 149.9881 | 150.6199 | 151.1731 | 151.7706 |
| 704 | 0 | 0 | 0 | 149.3794 | 149.8919 | 150.5363 | 151.1388 | 151.7605 |
| 705 | 0 | 0 | 0 | 149.0631 | 149.6055 | 150.2987 | 150.9426 | 151.6024 |
| 706 | 0 | 0 | 0 | 148.8545 | 149.4328 | 150.1623 | 150.8357 | 151.5076 |
| 707 | 0 | 0 | 0 | 148.5667 | 149.0889 | 149.8278 | 150.5064 | 151.2145 |
| 708 | 0 | 0 | 79.5865 | 148.4687 | 148.9668 | 149.6774 | 150.3398 | 151.0537 |
| 709 | 0 | 0 | 0 | 148.1245 | 148.6342 | 149.3427 | 149.9992 | 150.7398 |
| 710 | 0 | 0 | 0 | 147.8907 | 148.4478 | 149.1485 | 149.8327 | 150.5936 |
| 711 | 0 | 0 | 0 | 147.6852 | 148.2042 | 148.8487 | 149.5141 | 150.2887 |
| 712 | 0 | 0 | 147.1498 | 147.6198 | 148.1050 | 148.6900 | 149.3514 | 150.1184 |
| 713 | 0 | 0 | 0 | 147.4475 | 147.8533 | 148.3973 | 149.0470 | 149.8078 |
| 714 | 0 | 0 | 0 | 147.3518 | 147.7377 | 148.2740 | 148.9260 | 149.6638 |
| 715 | 0 | 0 | 0 | 146.8502 | 147.3183 | 147.9437 | 148.5941 | 149.3264 |
| 716 | 0 | 0 | 146.1400 | 78.0601 | 146.9446 | 147.6572 | 148.3572 | 149.1077 |
| 717 | 0 | 146.5547 | 77.5791 | 77.7490 | 78.2190 | 147.2639 | 147.9300 | 148.6881 |
| 718 | 0 | 0 | 78.0312 | 77.8129 | 78.0992 | 147.0248 | 147.6260 | 148.3788 |

Initial point

The point of the first region growth judgment

The point of the second region growth judgment

The points of the third region growth judgment

**Figure 7.** Schematic diagram of region growth.

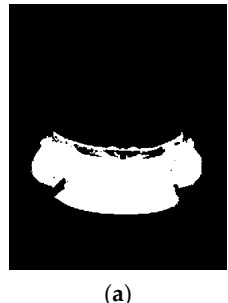

(a)

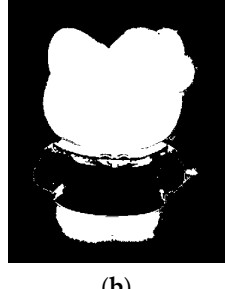

(b)

**Figure 8.** Object 1 point cloud mapping image segmentation diagram. (**a**) First segmented region. (**b**) The mapping image to be segmented.

Table 1 shows the results of region segmentation of point cloud mapping images through region growth and noise region judgment. In Table 1, there are a total of 95 regions after image segmentation. Among these 95 regions, there are three reference regions without noise, 66 noise regions, and 26 undetermined regions. Undetermined regions are regions that may contain noise. Table 2 shows the results of further judgment on the undetermined regions in Table 1 by using the KNN algorithm. According to Tables 1 and 2, there are 66 noise regions after image segmentation, and the undetermined region is judged by the k-nearest neighbor algorithm to obtain the undetermined There are 25 noise regions in the region. Therefore, a total of 81 noise regions were recorded. All the pixels in the 81 regions are indexed in the point cloud mapping image, and the pixel value of the pixel where the noise region is located is set to 0 to achieve the effect of noise removal. The 3D point cloud is restored according to the noise removal region, and the 3D point cloud model after noise removal is obtained, as shown in Figure 6d.

**Table 1.** Image segmentation result of object 1.

| Region Type | Number of Regions | Area |
|---|---|---|
| All regions | 95 | 189,121 |
| Noise-free reference regions | 3 | 169,811 |
| Noise regions | 66 | 3828 |
| Undetermined regions | 26 | 15,482 |

**Table 2.** Judgment result of undetermined regions of object 1.

| Reference Region Number | $Z_1$ | $Z_2$ | Euclidean Distance | Noise Region Judgment |
|---|---|---|---|---|
| 1 | 159.8183 | 215.2727 | 55.4544 | yes |
| 2 | 144.2411 | 75.3465 | 68.8946 | yes |
| 3 | 143.5596 | −4.5771 | 148.1367 | yes |
| 4 | 158.7262 | 92.7360 | 65.9902 | yes |
| 5 | 160.9971 | 14.6061 | 146.3910 | yes |
| 6 | 151.4172 | 208.5608 | 57.1436 | yes |
| 7 | 205.1798 | 147.1611 | 58.0187 | yes |
| 8 | 181.1901 | 118.2757 | 62.9144 | yes |
| 9 | 138.8656 | 197.0954 | 58.2298 | yes |
| 10 | 147.4264 | 80.3247 | 67.1017 | yes |
| 11 | 183.5244 | 184.7373 | 1.2129 | no |
| 12 | 167.6902 | 222.4362 | 54.7460 | yes |
| 13 | 147.9508 | 80.2411 | 67.7097 | yes |
| 14 | 150.0689 | 206.8956 | 56.8267 | yes |
| 15 | 156.9197 | 89.3048 | 67.6149 | yes |
| 16 | 184.3475 | 49.5676 | 134.7799 | yes |
| 17 | 161.3780 | 96.4765 | 64.9015 | yes |
| 18 | 181.5619 | 234.0803 | 52.5184 | yes |
| 19 | 152.5448 | 208.5010 | 55.9562 | yes |
| 20 | 172.0235 | 225.6556 | 53.6321 | yes |
| 21 | 151.5831 | 207.7492 | 56.1661 | yes |
| 22 | 153.0938 | 6.4802 | 146.6136 | yes |
| 23 | 155.6365 | 88.3370 | 67.2995 | yes |
| 24 | 212.8726 | 156.0542 | 56.8184 | yes |
| 25 | 153.3561 | 209.5078 | 56.1517 | yes |
| 26 | 142.6327 | 94.4186 | 48.2141 | yes |

According to the position index of the denoising point cloud on the three-dimensional point number mapping image in Figure 6d, the binary image of the point cloud mapping of the noise removal regions is established, as shown in Figure 9a. In Figure 9a, only the values of the pixels containing the point cloud are set to 1. Figure 9b is the binary image of the point cloud map image in Figure 5d. The point cloud map image after denoising

in Figure 9a and the point cloud map image without denoising in Figure 9b is calculated according to the restored point cloud in Section 3.5 outside the regions of the noise-free point cloud, and The XOR operation is performed in the three noise-free point cloud regions in Table 1, and the mapping image of the restored regions in Figure 9c is obtained. Phase recovery is carried out according to the recovery regions, and the recovered absolute phase information is obtained. According to Figure 1, combined with the original 3D calibration parameters, 3D reconstruction is performed on the new absolute phase image to obtain a noise-free 3D point cloud. Figure 10a is the original image of part of the absolute phase, and Figure 10b is the restored absolute phase image; it can be concluded that after the noise regions are deleted and restored according to the reference phase, the restored absolute phase information is more accurate than the original image, no new point cloud noise will be generated. Figure 11 is a noise-free point cloud image obtained after a three-dimensional reconstruction of the restored absolute phase image.

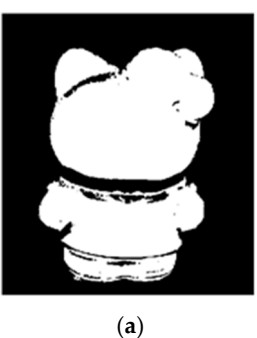 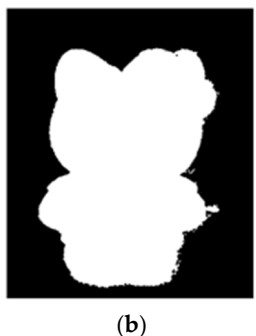 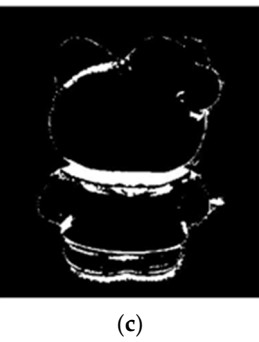

(a) (b) (c)

**Figure 9.** Point cloud mapping binary image of object 1. (**a**) Denoised point cloud mapping image. (**b**) Initial point cloud mapping image. (**c**) Recovery point cloud mapping image.

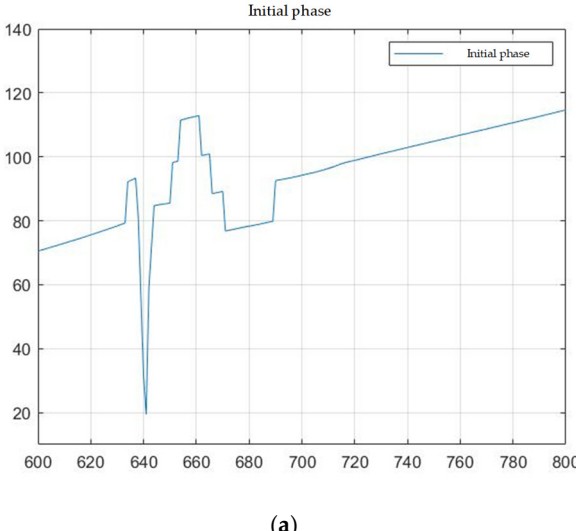 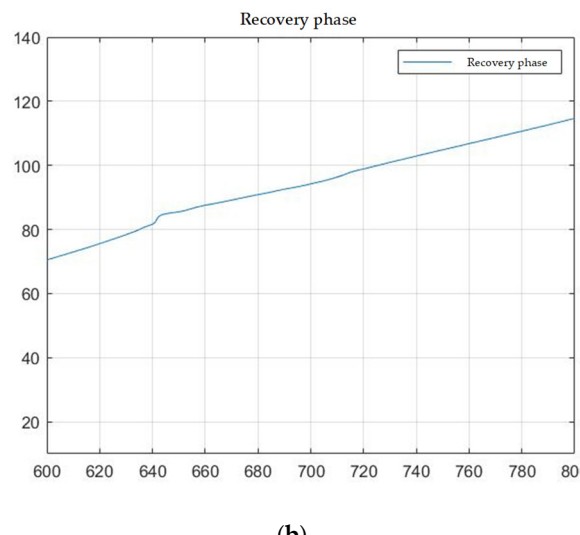

(a) (b)

**Figure 10.** Absolute phase image of part of object 1. (**a**) Partial absolute phase image of initial point cloud. (**b**) Partial absolute phase image of recovery point cloud.

In order to verify the robustness of the proposed method, the proposed method is compared with the radius filtering algorithm [35] and voxel filtering algorithm [36] commonly used in image processing, as well as the point cloud filter algorithm proposed in [37], and the point cloud denoising results are compared and analyzed. Table 3 is the comparative analysis of point cloud denoising results of object 1. According to Table 3, it can be seen that the fastest speed of the voxel filtering algorithm in the comparative analysis is only 0.081 s, but $P_2$ is higher than other methods and $P_1$ is lower than other

methods. The algorithm does not have a high degree of discrimination for noise areas, and the accuracy of point clouds is not high, only 13.977%. The radius filtering algorithm is better, the point cloud denoising accuracy can reach 89.314%, and the required time is only 0.755 s, $P_2$ is larger than the proposed method and the algorithm proposed in the literature [37]. Compared with the literature [37], the proposed method uses a region growing and image segmentation to replace the image processing part in literature [37] to achieve 3D point cloud denoising, and the proposed method has higher point cloud accuracy, which can reach 99.974%, The point cloud denoising time is 0.954 s.

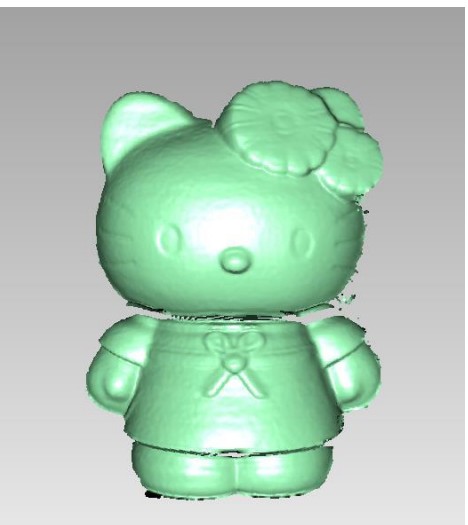

**Figure 11.** 3D point cloud model without noise of object 1.

**Table 3.** Comparison analysis of point cloud denoising results of object 1.

| Denoising Calculation Parameters | Ours | Radius Filtering Algorithm | Voxel Filtering Algorithm | The Algorithm Proposed in [37] |
|---|---|---|---|---|
| Number of initial point clouds (pixel) | 169,854 | 187,940 | 90,878 | 167,795 |
| point cloud denoising time (s) | 0.954 | 0.755 | 0.081 | 0.909 |
| $P_1$ (pixel) | 169,811 | 167,857 | 12,702 | 167,736 |
| $P_2$ (pixel) | 43 | 20,083 | 78,176 | 60 |
| $Q$ (%) | 99.974 | 89.314 | 13.977 | 99.964 |

*3.4. Point Cloud Denoising Experiment 2*

In order to verify the robustness of the proposed method, a Rubber Duck doll is selected as object 2 to be measured in the three-dimensional reconstruction experiment. The experimental process of 3D reconstruction of object 2 is shown in Figure 12. It can also be known from Figure 12d that the 3D point cloud model of object 2 also has point cloud noise. It is also necessary to perform a point cloud denoising on object 2.

A point cloud denoising experiment is carried out on object 2, and a comparison analysis is carried out with other denoising algorithms mentioned above. In the point cloud denoising experiment, the selected image segmentation parameters, including the selection of region growth seed points, growth step size, growth window, color feature extraction method, region growth threshold $\omega$, and Euclidean distance threshold $\delta$ remain unchanged. Figure 13 is the experimental image of the point cloud denoising method based on image segmentation. Tables 4 and 5, respectively, show the results of image segmentation of the point cloud mapping image of object 2 and the results of further judging whether all undetermined regions are noise regions according to the above steps. In Table 4, object 2 is divided into 22 regions in total, including 1 reference region, 9 noise regions, and 12 undetermined regions. In Table 5, it is judged that there are 11 undetermined area noise

regions. According to Tables 4 and 5, all recorded noises are removed from the point cloud mapping image in Figure 13b, and the point cloud mapping image of the noise-removed area is obtained, as shown in Figure 13c.

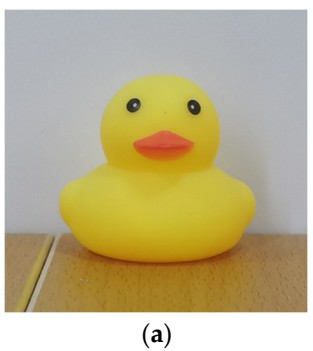 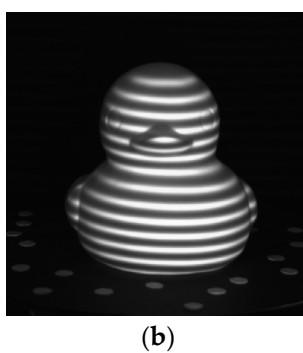 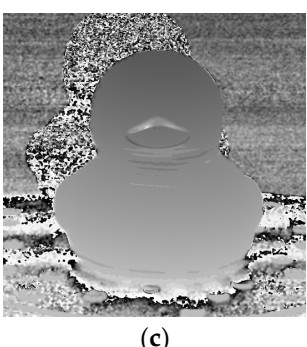 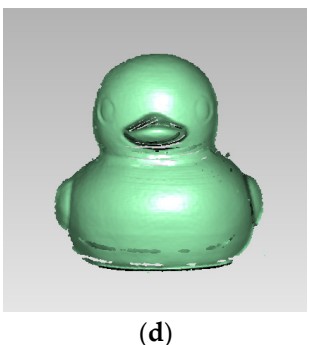

(**a**)   (**b**)   (**c**)   (**d**)

**Figure 12.** Three-dimensional reconstruction experiment image of Object 2. (**a**) Experimental object 2. (**b**) Deformed fringe image. (**c**) Absolute phase pattern. (**d**) Three-dimensional point cloud of the object 2.

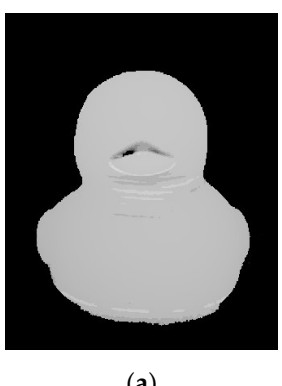 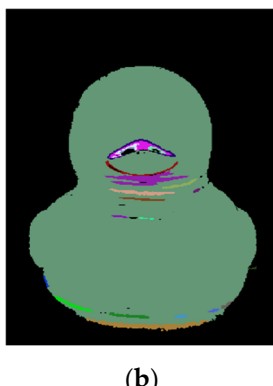 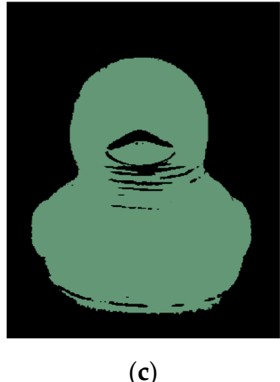 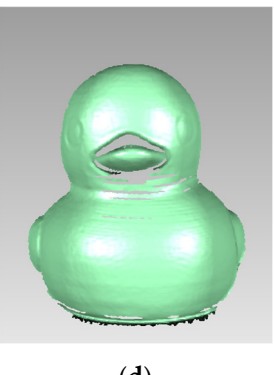

(**a**)   (**b**)   (**c**)   (**d**)

**Figure 13.** Experimental image of point cloud denoising method based on image segmentation of Object 2. (**a**) Point cloud mapping image. (**b**) Segmented mapping image. (**c**) Denoised image. (**d**) Denoised 3D point cloud.

**Table 4.** Image segmentation result of object 2.

| Region Type | Number of Regions | Area |
| --- | --- | --- |
| All regions | 22 | 120,613 |
| Noise-free reference regions | 1 | 112,871 |
| Noise regions | 9 | 1423 |
| Undetermined regions | 12 | 6319 |

According to the denoised 3D point cloud of object 2 and the creation of the point cloud mapping image, a binary point cloud mapping image is generated, as shown in Figure 14. The corresponding operations are then applied to the mapped images in Figure 14a,b for point cloud restoration, resulting in the recovery point cloud mapping image depicted in Figure 14c. The absolute phase information is restored based on the recovery region in Figure 14c, resulting in Figure 15, which displays the absolute phase image of part 2 of the object. By denoising the initial absolute phase using the point cloud, an absolute reference phase is established through the coordinate points of the noise region, and accurate restored absolute phase information is obtained through calculation. As depicted in Figure 15b, the recovered absolute phase appears smoother without the occurrence of sharp phase changes seen in Figure 15a. Figure 16 illustrates a noise-free point cloud image obtained through a three-dimensional reconstruction of the restored absolute phase image.

**Table 5.** Judgment result of undetermined regions of object 2.

| Reference Region Number | $Z_1$ | $Z_2$ | Euclidean Distance | Noise Region Judgment |
|---|---|---|---|---|
| 1 | 93.7989 | −4.9949 | 98.7938 | yes |
| 2 | 63.8626 | 176.2533 | 112.3907 | yes |
| 3 | 5.1033 | 106.9922 | 101.8889 | yes |
| 4 | 5.9464 | 88.0945 | 82.1481 | yes |
| 5 | −11.7774 | 85.5615 | 97.3389 | yes |
| 6 | −12.0302 | 72.3592 | 84.3894 | yes |
| 7 | 82.3165 | 203.8977 | 121.5812 | yes |
| 8 | 126.7995 | 35.2944 | 91.5051 | yes |
| 9 | 200.5824 | 356.9985 | 156.4161 | yes |
| 10 | 79.3232 | 74.3904 | 4.9328 | no |
| 11 | 178.7573 | 329.0114 | 150.2541 | yes |
| 12 | −0.4411 | 81.4749 | 81.9160 | yes |

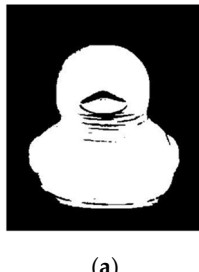

(**a**)

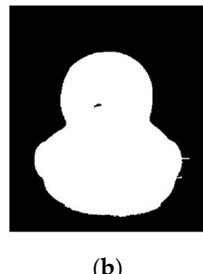

(**b**)

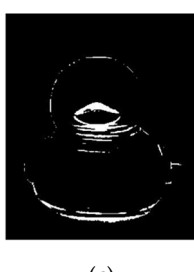

(**c**)

**Figure 14.** Point cloud mapping binary image of object 2. (**a**) Denoised point cloud mapping image. (**b**) Initial point cloud mapping image. (**c**) Recovery point cloud mapping image.

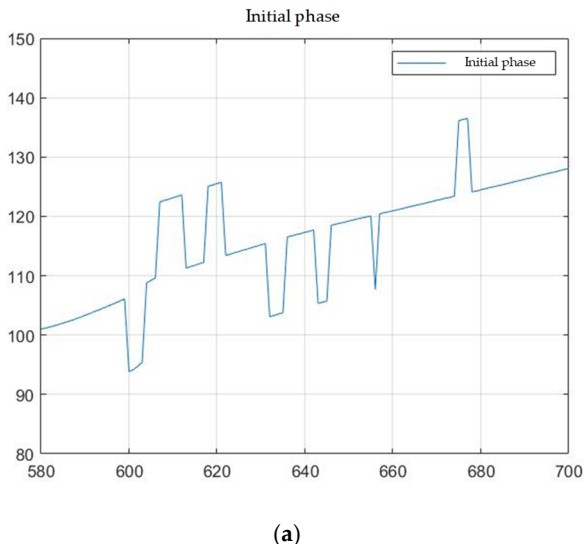

(**a**)

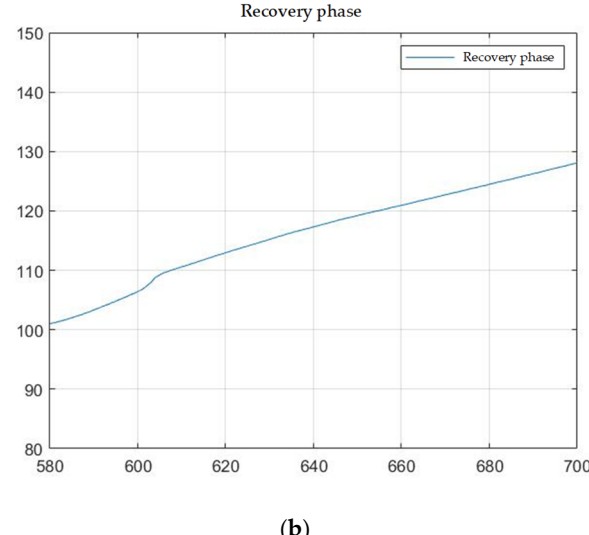

(**b**)

**Figure 15.** Absolute phase image of part of object 2. (**a**) Partial absolute phase image of initial point cloud. (**b**) Partial absolute phase image of recovery point cloud.

Table 6 is the comparative analysis of point cloud denoising results of object 2. According to Table 6, the point cloud denoising accuracy Q of the proposed method is the highest, which is 99.944%, and the denoising time is 0.875 s.

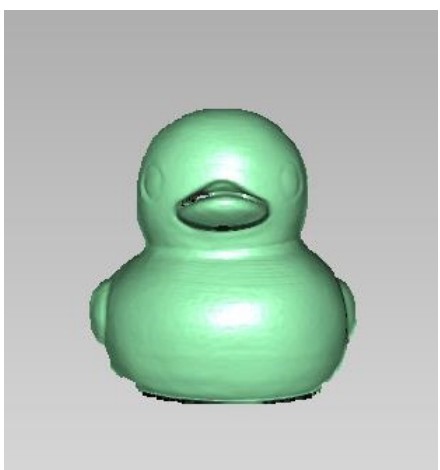

**Figure 16.** 3D point cloud model without noise of object 2.

**Table 6.** Comparison analysis of point cloud denoising results of object 2.

| Denoising Calculation Parameters | Ours | Radius Filtering Algorithm | Voxel Filtering Algorithm | The Algorithm Proposed in [37] |
|---|---|---|---|---|
| Number of initial point clouds (pixel) | 112,934 | 124,960 | 60,432 | 111,565 |
| point cloud denoising time (s) | 0.875 | 0.574 | 0.058 | 0.862 |
| $P_1$ (pixel) | 112,871 | 119,174 | 10,309 | 111,495 |
| $P_2$ (pixel) | 63 | 5786 | 50,123 | 70 |
| $Q$ (%) | 99.944 | 95.370 | 17.059 | 99.937 |

According to the point cloud experiments, the proposed method employs image segmentation to quickly identify and remove scattered and blocky noise points in the 3D point cloud reconstructed using fringe projection. The denoised point cloud, obtained through absolute phase recovery, improves the accuracy of denoising. The entire denoising process takes less than 1 s, demonstrating real-time performance. However, the noise-free 3D point cloud model exhibits vacancies due to the coverage during the fringe projection process, leading to the loss of 3D information in those regions. Future research will focus on combining the proposed method with 3D point cloud registration technology to enhance its practicality.

*3.5. Parameter Selection Comparison*

In this section, we compare the performance of our proposed algorithm under different parameter settings to demonstrate that our chosen parameters are optimal. We will compare the following key parameters: the region growth threshold $\omega$ and the Euclidean distance threshold $\delta$ in the KNN algorithm.

The performance of the proposed method under different region growth thresholds and Euclidean distance thresholds is evaluated. We set different region growth thresholds: 0–100 and experimented on the same objects. The parameter comparison results are shown in Figure 17. We set different Euclidean distance thresholds: 0–100 and experimented on the same objects. The parameter comparison results are shown in Figure 18.

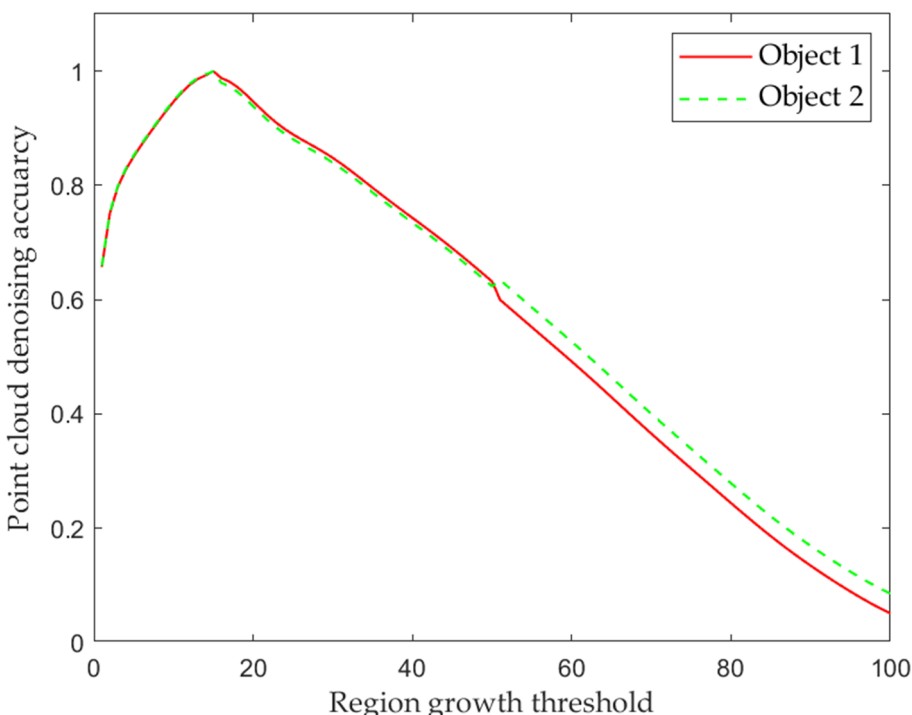

**Figure 17.** Region growth threshold comparison results.

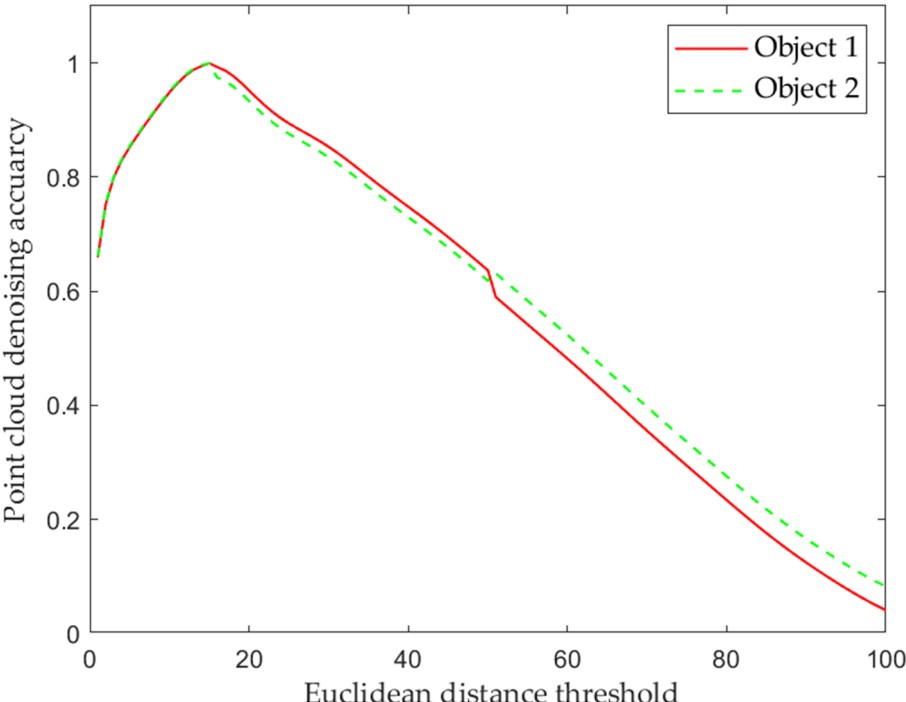

**Figure 18.** Euclidean distance threshold comparison results.

The experimental results show that when the region growth threshold is $\omega = 15$, and the Euclidean distance threshold is $\delta = 15$, the proposed method achieves the best performance index and has the highest point cloud denoising accuracy. A lower region growth threshold or Euclidean distance threshold results in a lot of segmented regions, and the removal of too many noise regions leads to lower accuracy of point cloud denoising. A higher region growth threshold or Euclidean distance threshold will result in very

few segmented regions, resulting in the situation that some point cloud noise cannot be removed, reducing the accuracy of point cloud denoising.

According to the above comparative experiments, we determined that the region growth threshold is $\omega = 15$, and the Euclidean distance threshold is $\delta = 15$. These parameters are set in the point cloud data centers of the two objects in the point cloud denoising experiment to achieve the best performance with the highest point cloud denoising accuracy. The experimental results show that the parameters we choose have an important impact on the performance of the algorithm and prove its rationality and effectiveness in our algorithm.

## 4. Conclusions

In this paper, a point cloud filtering method for 3D imaging obtained by fringe projection is proposed. To process the point cloud, a point cloud mapping image is established and segmented, and the judged noise regions are removed from the image. Then, the 3D point cloud is mapped to the 2D image for calculation and judgment, which avoids performing complex calculations in three dimensions. Experiments have proved that this method can effectively remove various noise point clouds, and the proposed method in this paper can restore part of the noise point cloud to a noise-free point cloud and improve the overall accuracy of the 3D point cloud. The point cloud denoising accuracy can reach up to 99.974% after the proposed method is used for point cloud denoising. For a 3D point cloud with no more than 160,000 pixels, the calculation time for denoising is less than 1 s in our experimental platform. The proposed method is suitable for point cloud processing on the 3D imaging result obtained by digital fringe projection.

**Author Contributions:** Conceptualization, J.Z. and Y.D.; methodology, J.Z., J.L., Z.D. and Y.D.; software, J.Z. and Z.D.; validation, J.Z., J.L. and Y.D.; formal analysis, J.Z., J.L., Y.D., Z.D. and J.X.; investigation, J.Z., J.L., Z.D. and Y.D.; resources, J.Z., Z.D. and J.X.; data curation, J.Z. and J.L.; writing—original draft preparation, J.Z. and Z.D.; writing—review and editing, J.Z. and Y.D.; visualization, J.Z. and Y.D.; supervision, Y.D. and J.X.; project administration, Y.D.; funding acquisition, J.L., Y.P. and J.X. All authors have read and agreed to the published version of the manuscript.

**Funding:** This work was supported by the Key Scientific Research Platforms and Projects of Ordinary Universities in Guangdong Province (Grant No. 2021KCXTD051), the Wuyi University Hong Kong and Macau Joint R&D Fund Project (Grant No. 2021WGALH17) and the Key Project of Basic and Applied Basic of Jiangmen City (Grant No. 20210301037300007331).

**Institutional Review Board Statement:** Not applicable.

**Informed Consent Statement:** Not applicable.

**Data Availability Statement:** The data presented in this study are available on reasonable request from the corresponding author.

**Conflicts of Interest:** The authors declare no conflict of interest.

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
