# Peer review of "Three-Dimensional Point Cloud-Filtering Method Based on Image Segmentation and Absolute Phase Recovery"

_electronics, doi:10.3390/electronics12122749_

Round 1

Reviewer 1 Report

This manuscript proposes a point cloud denoising method based on segmentation. The writing is well-organized but the novelty is very limited. The authors should carefully revise the manuscript. Please kindly find my concerns as follows:

1. The motivation is lacking. The authors should detail analyze the current methods. The abstract, "The existing point cloud filtering methods involve analyzing the geometric relationship between 3D space and point cloud, which poses challenges such as complex calculation and low efficiency", is meaningless to me. Also, the introduction is insufficient (from line 67), why did the authors propose it?

2. Lacks ablation studies. Since many technologies are from previous papers with marginal novelty, e.g. KNN and segmentation, I think the author at least should test the parameters and show the effectiveness o the selected modules.

3. Experiments: From Figure 9, I can see many vacancies in the restored regions.  The authors should analyze the reason and show more comparisons. By the way, Figure 5 is very blurry, I can not see 5 (d) clearly. Also, the authors are suggested to visualize the difference between original point clouds and restored point clouds, i.e., quantitatively comparison between 5 (d) with 6 (d) and 9.

4. Also, only one object is shown in the experiment. I think the authors should share more results, to show the robustness.

5. Lacks some introduction to other 3D reconstruction methods, e.g. photometric stereo [1,2] and binocular stereo[3,4].

[1] Non-lambertian photometric stereo network based on inverse reflectance model with collocated light

[2] Recovering surface normal and arbitrary images: A dual regression network for photometric stereo

[3] High quality 3D reconstruction based on fusion of polarization imaging and binocular stereo vision

[4] Deep-learning assisted high-resolution binocular stereo depth reconstruction

6. Some typos: such as line 234 and line 270.

Author Response

This manuscript proposes a point cloud denoising method based on segmentation. The writing is well-organized but the novelty is very limited. The authors should carefully revise the manuscript. Please kindly find my concerns as follows:

[Comment 1]

The motivation is lacking. The authors should detail analyze the current methods. The abstract, "The existing point cloud filtering methods involve analyzing the geometric relationship between 3D space and point cloud, which poses challenges such as complex calculation and low efficiency", is meaningless to me. Also, the introduction is insufficient (from line 67), why did the authors propose it?

[Response 1]

Thank you for the comments on our manuscript. In response to your concerns, we have made corresponding modifications and improvements. We have clarified the motivation of this manuscript as improving the point cloud quality and efficiency obtained by fringe projection 3D imaging, and expanded the Introduction on the major methods of point cloud denoising. According to the problems of existing point cloud denoising methods, we propose our method by exploiting the properties of region segmentation and absolute phase to remove the interference of noise on point cloud data. It makes the method more flexible and applicable in practical applications.

[Comment 2]

Lacks ablation studies. Since many technologies are from previous papers with marginal novelty, e.g. KNN and segmentation, I think the author at least should test the parameters and show the effectiveness of the selected modules.

[Response 2]

We have added an ablation study section in Section 3.5. We test and analyze the key parameters of image segmentation and undetermined region analysis, demonstrating the effectiveness of the selected modules.

[Comment 3]

Experiments: From Figure 9, I can see many vacancies in the restored regions. The authors should analyze the reason and show more comparisons. By the way, Figure 5 is very blurry, I cannot see 5 (d) clearly. Also, the authors are suggested to visualize the difference between original point clouds and restored point clouds, i.e., quantitatively comparison between 5 (d) with 6 (d) and 9.

[Response 3]

In our method, the vacancies in the 3D model of Figure 12 (original Figure 9) are resulted from the geometry configuration of fringe projection profilometry, which can be regarded as shadow area. The vacant area does not receive the projection of the light source, resulting in the loss of 3D information. We clarify this issue in the revised manuscript that the shadow regions do not receive fringe projections from light sources, therefore the vacant areas appear. Additionally, we have made Figure 5 more clear. We compare the difference between original and restored point clouds in terms of absolute phase maps and point cloud denoising accuracy in Figure 11 and Table 3.

[Comment 4]

Also, only one object is shown in the experiment. I think the authors should share more results, to show the robustness.

[Response 4]

To demonstrate the robustness of the method, we have added the experimental results in the revised manuscript with Hell Kitty doll and the Rubber Duck doll to verify the performance of the proposed method.

[Comment 5]

Lacks some introduction to other 3D reconstruction methods, e.g. photometric stereo [1,2] and binocular stereo[3,4].

[1] Non-lambertian photometric stereo network based on inverse reflectance model with collocated light

[2] Recovering surface normal and arbitrary images: A dual regression network for photometric stereo

[3] High quality 3D reconstruction based on fusion of polarization imaging and binocular stereo vision

[4] Deep-learning assisted high-resolution binocular stereo depth reconstruction

[Response 5]

We have introduced other 3D reconstruction methods and added these references into the revised manuscript.

[Comment 6]

Some typos: such as line 234 and line 270.

[Response 6]

We carefully checked the paper for spelling errors and made corrections.

Reviewer 2 Report

Interesting topic of work. The presented method is promising (certain elements of this method can be found in commercial programs for reconstruction of 3D images from 2D images - e.g. Materialize Mimics). The method still requires a lot of work, because the presented effects are not as great as the authors claim.

The work requires a thorough construction of the Conclusions section. This section is poor in information.

The literature review also needs to be supplemented - a cursory review of WoS or Scopus shows a great interest of scientists in this topic. In addition, the literature review is geographically limited, out of 20 literature items, as many as 13 are authored only by scientists from one circle of opinions and views.

the language layer of the publication is correct

Author Response

[Comment 1]

Interesting topic of work. The presented method is promising (certain elements of this method can be found in commercial programs for reconstruction of 3D images from 2D images - e.g. Materialize Mimics). The method still requires a lot of work, because the presented effects are not as great as the authors claim.

The work requires a thorough construction of the Conclusions section. This section is poor in information.

The literature review also needs to be supplemented - a cursory review of WoS or Scopus shows a great interest of scientists in this topic. In addition, the literature review is geographically limited, out of 20 literature items, as many as 13 are authored only by scientists from one circle of opinions and views.

[Response 1]

Thank you very much for your valuable comments. To indicate the performance of our proposed methods, we have added point cloud denoising accuracy and calculation time of point cloud denoising in the revised manuscript. In Section 4 of the revised manuscript, we elaborate the calculation process of the region judgment and give the calculation parameters of the proposed method. In Section 4.4 of the revised manuscript, we conduct comparative experiments on key parameters to demonstrate the effectiveness of the selected parameters.

We also provide more backgrounds on recent research and the current state of the field in Introduction. We have included references to recently published review papers in the revised manuscript and cover more relevant research, making sure it provides sufficient background information and citations to the papers. In Introduction of the revised manuscript, we analyze existing point cloud denoising methods and their shortcomings. According to existing point cloud denoising problems, we propose the background and motivation of our method.

Reviewer 3 Report

This paper describe a 3D piont cloud filtering method based on imaging segmentation and phase recovery. It can be taken as complementary work for filtering based denoising method for 3D reconstruction. Besides few minor issues listed below, the overall quality of the paper is fine for publication.

1, In abstract, the author claim their target is to overcome the issue of complex calculation and low efficiency and improve the accurcy and speed of 3D cloud generation. The experimental data is able to support the improvement of accuracy (Although it is only from single objective, more samples test may be considered to solid the conclusion). However, there is no analysis or discussion about the efficiency and speed of the proposed method, the author should prvoide those data e.g. speed, and benchmark with literature to support their claim. 

2. The introduction not provide enough background of the research such as state of art in this area. There is quite number publications should be included in the discussion or benchmarked with proposed method if possible. They can be found in a review paper published recently :

Lang Zhou, Guoxing Sun, Yong Li, Weiqing Li, Zhiyong Su, "Point cloud denoising review: from classical to deep learning-based approaches", Graphical Models, Volume 121, 2022, 101140 The correct link is below https://www.sciencedirect.com/science/article/abs/pii/S1524070322000170?via%3Dihub

Additional comments

1. What is the main question addressed by the research?

This paper provides a filtering based solution for 3D point cloud denoising.

2. Do you consider the topic original or relevant in the field? Does it address a specific gap in the field?

The topic is relevant in the field. It maybe can be considered as complementary or potential approach to improve the performance of the denoising technique. However, the author doesn't provide data on the calculation workload or speed test as well as how the proposed method will behave when it is benchmarked with other approaches reported in literature. It is hard to judge whether it is able to address the efficiency, complexity or speed issue mentioned in the introduction section of the manuscript. That is why in my last comments, I suggest the author should execute a comparison study to support their claim. Otherwise, they should revise their claim.

3. What does it add to the subject area compared with other published material?

The author did not provide any data in the manuscript to evaluate the performance of the method they proposed. The author converts the 3D problem to 2D image processing, which is a very matured technique. Some speed improvement can be expected. But there is no data on how fast the proposed method in current manuscript.

4. What specific improvements should the authors consider regarding the methodology? What further controls should be considered?

The major weakness of the work is the generalization of proposed method is questionable as that its efficiency was only tested using single object. The proposed method should be tested with more objects. Another issue, the author should consider benchmark with other filtering based denoising methods published in literature (They can find it in the paper link I attached in question 8).

5. Are the conclusions consistent with the evidence and arguments presented and do they address the main question posed?

Actually, author claimed different things in introduction and conclusion sections. In the introduction, they claim they will overcome the efficiency and complexity issue of denoising algorithm. I can not agree with it as there is no efficiency analysis or speed evaluation in the paper. But in conclusion, their claim a method can "restore ... to a noise-free point cloud" . It is partially true and may have impact in this area. However, authors need to either revise their conclusion carefully or provide more data to support all their claims.

6. Are the references appropriate?

The reference is fine for me.

7. Please include any additional comments on the tables and figures.

I am fine with the table and figures except some minor inconsistence of name used throughout the paper.

The english writing is fine for me.

Author Response

This paper describes a 3D point cloud filtering method based on imaging segmentation and phase recovery. It can be taken as complementary work for filtering based denoising method for 3D reconstruction. Besides few minor issues listed below, the overall quality of the paper is fine for publication.

[Comment 1]

In abstract, the author claim their target is to overcome the issue of complex calculation and low efficiency and improve the accuracy and speed of 3D cloud generation. The experimental data is able to support the improvement of accuracy (Although it is only from single objective, more samples test may be considered to solid the conclusion). However, there is no analysis or discussion about the efficiency and speed of the proposed method, the author should provide those data e.g. speed, and benchmark with literature to support their claim.

[Response 1]

We appreciate your pointing out the efficiency and speed of our approach. In the revised manuscript, we provide data on the calculation time and accuracy of point cloud denoising, the results are also compared with other methods. Based on the comparative experiments, it can be concluded that our proposed method has high accuracy and fast speed. These results provide a more comprehensive evaluation of our method.

[Comment 2]

The introduction not provide enough background of the research such as state of art in this area. There is quite number publications should be included in the discussion or benchmarked with proposed method if possible. They can be found in a review paper published recently :

Lang Zhou, Guoxing Sun, Yong Li, Weiqing Li, Zhiyong Su, "Point cloud denoising review: from classical to deep learning-based approaches", Graphical Models, Volume 121, 2022, 101140 The correct link is below:

https://www.sciencedirect.com/science/article/abs/pii/S1524070322000170?via%3Dihub.

[Response 2]

We appreciate your suggestion on the literature review. We have conducted a thorough review of recent publications in point cloud denoising, including the review paper you mentioned. We have included references to relevant works and discussed their contributions in our revised manuscript. This addition will enhance the context and provide a broad perspective on existing research in the field.

Round 2

Reviewer 1 Report

The authors have addressed all my concerns. Thank you.